# Data-Mining Techniques Based Relaying Support for Symmetric-Monopolar-Multi-Terminal VSC-HVDC System

**Abha Pragati [1], Debadatta Amaresh Gadanayak [2] , Tanmoy Parida [1] and Manohar Mishra [2],\***

1   Department of Electrical Engineering, ITER, Siksha O Anusandhan University, Bhubaneswar 751030, India
2   Department of Electrical and Electronics Engineering, ITER, Siksha O Anusandhan University, Bhubaneswar 751030, India
\*   Correspondence: manoharmishra@soa.ac.in

**Abstract:** Considering the advantage of the ability of data-mining techniques (DMTs) to detect and classify patterns, this paper explores their applicability for the protection of voltage source converter-based high voltage direct current (VSC-HVDC) transmission systems. In spite of the location of fault occurring points such as external/internal, rectifier-substation/inverter-substation, and positive/negative pole of the DC line, the stated approach is capable of accurate fault detection, classification, and location. Initially, the local voltage and current measurements at one end of the HVDC system are used in this work to extract the feature vector. Once the feature vector is retrieved, the DMTs are trained and tested to identify the fault types (internal DC faults, external AC faults, and external DC faults) and fault location in the particular feeder. In the data-mining framework, several state-of-the-art machine learning (ML) models along with one advanced deep learning (DL) model are used for training and testing. The proposed VSC-HVDC relaying system is comprehensively tested on a symmetric-monopolar-multi-terminal VSC-HVDC system and presents heartening results in diverse operating conditions. The results show that the studied deep belief network (DBN) based DL model performs better compared with other ML models in both fault classification and location. The accuracy of fault classification of the DBN is found to be 98.9% in the noiseless condition and 91.8% in the 20 dB noisy condition. Similarly, the DBN-based DMT is found to be effective in fault locations in the HVDC system with a smaller percentage of errors as MSE: 2.116, RMSE: 1.4531, and MAPE: 2.7047. This approach can be used as an effective low-cost relaying support tool for the VSC-HVDC system, as it does not necessitate a communication channel.

**Keywords:** HVDC; fault detection; fault location; data-mining; machine learning; deep learning; voltage source converter

## 1. Introduction

### 1.1. Motivation and Incitement

For long distances and huge amounts of power transfer, HVDC technology has proven to be the most efficient, cost-effective, and established technology. There are several technological and environmental advantages of the HVDC system compared with HVAC, such as less transmission loss, lesser required transmission tower, improved system stability, narrowed transmission corridor, higher efficiency, better power quality, and being eco-friendly [1–3]. However, HVDC systems face various challenges in terms of protection viewpoints, more specifically relating to fault detection, classification, and location [3]. The DC fault current has a large peak and steady values within a few milliseconds. This requires high-speed fault detection and isolation methods to handle the critical implications in an HVDC grid. In this regard, intelligent fault protection schemes are required against the chance of damage triggered by short circuit faults and to provide continuous power to connected loads. The schemes should pose self-diagnosing and self-healing capabilities that can detect and isolate the faulty section as soon as possible [3].

*1.2. Literature Review*

With the intention of safe operation of HVDC systems, a quick action related to fault identification and tripping circuit breaker is essential [4]. For this purpose, several protection schemes were already proposed in much of the literature. These schemes can be categorized as one-end or multi-ends measurement-based HVDC fault detection schemes.

The following are a few examples of multi-ends measurement-based HVDC fault detection schemes: the ratio of transient voltage and current at both ends [5], differential protection based on the compensation of distributed capacitive currents at both ends [6], differential protection based on the value and polarity characteristics of the subtracted both-ends-synchronized current signal, traveling wave (TW) based protection [7], directional protection [8], analyzing transient harmonic currents at both ends [9], and wavelet transform based protection [10–12]. Similar to fault detection, some examples of the HVDC transmission line's fault location schemes based on multi-end measurements are as follows: TW-based schemes [13–17] and a few signal processing (SP) based schemes such as discrete wavelet transform [18–20], s-transform [21], Fourier transforms [22], mathematical morphology [23], and empirical mode decomposition [24]. Here, the TW-based approach has been a well-known fault locating technique for an HVAC system for the last couple of decades. In TW-based schemes, high-speed protection can only be possible with the use of the first incident traveling wave, and the accuracy of fault location can be hampered in the case of very long transmission lines. Moreover, the computation burden, system complexity, and additional processing time involved in SP-based schemes may delay the switching of the DC circuit breaker. In addition to these disadvantages, the multi-end measuring approach faces delays due to the requirement of data transmission, and the overall performance of the system depends on the reliability of communication channels [25]. As the one-end measurement-based schemes are less costly, as well as having less selectivity as compared with multi-end information-based methods, this work proposed a fault detection and location scheme based on single-end measurements.

Analysis of single-end measurement-based TW schemes was reported in the literature [26–28]. However, the requirement of a large sampling rate for data extraction so as to correctly acquire the wave front arrival time is a major disadvantage of this approach [29]. In contrast to this, several authors have proposed one-end measuring-based schemes for detecting faults in the HVDC system. Kong et al. [30] proposed a protection scheme for the HVDC system using rectifier side voltage and current signals. The scheme comprised five major units such as a detection unit, boundary unit, directional unit, faulted pole identification unit, and lightning disturbance identification unit. Suonan et al. [31] presented a protection scheme for an HVDC system using frequency-dependent parameters extracted from voltage and current signals measured at one-end. Similarly, the following are the methods used for the protection of HVDC lines using one-end local measures: distance protection based schemes [32], directional and boundary element based method [33,34], Teager energy operator based schemes [35], short-term Fourier transforms [36], wavelet transform based schemes [26,37,38], s-transform based schemes [39], and Hilbert–Huang transform based schemes [40]. However, in these types of schemes, an extensive simulation or analysis is needed to set the thresholds. Providing proper thresholding is a quite difficult task and therefore may be unsuitable to provide a stand-alone protective solution.

Currently, data mining (DM) and machine learning (ML) based artificial intelligence (AI) methods have gained huge attention for fault detection, classification, and location in HVAC systems [41–45], as they are known to be the most effective tools for solving complex problems. However, the application of the data-mining approach in the HVDC system has seen limited study in the past literature. In this regard, the artificial neural network (ANN) is one of the most popularly used AI methods in the protection of HVDC systems [46–51]. In addition to this, some other AI methods for application in HVDC protection lines are as follows: a fuzzy logic theory based approach [52], a support vector machine (SVM) based approach [26,53], an extreme learning machine [54], and a stacked autoencoder [55]. The ML-based fault detection schemes pose a fast execution period and have good sensitivity

and better reliability. Therefore, this work has tried to explore the applicability of different data-mining techniques in HVDC fault detection, classification, and location.

*1.3. Major Contributions and Organization*

This paper presents an effective approach based on data-mining (DM) techniques (DMTs) for the protection of the HVDC system. As a whole, the work concentrates on the detection and classification of both internal and external faults in a multi-terminal VSC-HVDC system. The internal faults include positive pole-to-ground, negative pole-to-ground, and pole-to-pole faults that occurred in a DC transmission line, whereas the external faults are either DC faults that occurred other than at the measured DC-transmission line or AC faults encountered at either rectifier-end/inverter-end.

The general objective of the proposed work is to analyze the applicability of different data-mining techniques in HVDC fault detection and location processes. In this regard, the major contributions of the article are summarized as follows:

- Initially, the DC-voltage and DC-current signals are retrieved at the relay terminal of the studied HVDC system, which are later used to extract several sensitive features.
- Afterward, these features are used to build the data-mining model-based black-box solution for reporting the faults level and faults distance by ensuring the tripping command.
- In the DM framework, ML-based techniques such as random forest (RF), support vector machine (SVM), extreme learning machine (ELM), k-nearest neighbor (KNN), and RF are trained and tested using the extracted features.
- In order to increase the efficiency of the DM model by reducing the computational burden, the sequential forward feature selection (SFS) is integrated with each DM model.
- In addition to the ML technique, a deep belief network (DBN) based deep learning (DL) model is also trained and tested with the extracted features for fault detection and location in the HVDC system.
- The accuracy of each model is also tested with extremely noisy conditions.
- This approach can be used as an effective low-cost relaying support tool for the VSC-HVDC system, as it does not necessitate a communication channel.
- The proposed fault detection and location approach is comprehensively tested on a MATLAB simulation model (symmetric-monopolar VSC-HVDC system) and portrays promising results in various operating conditions.

The rest of the paper is organized as follows: Section 2 provides the details of the studied HVDC system. Section 3 portrays the proposed approach to detect and classify the HVDC faults. Section 4 provides a brief description of each data-mining model used in this study. Simulation results are presented in Section 5. Finally, Section 6 summarizes the article with several concluding remarks.

## 2. Studied HVDC System

To test the protection strategy, a four terminal VSC-HVDC system connecting 230 kV, 2000 MVA AC systems each, was implemented in MATLAB-SIMULINK (refer to Figure 1). The converter stations were interconnected through four ±100 kV, 100 MVA symmetrical monopolar pi lines to create a meshed system, as shown in Figure 1. The converters were fed from 250 MVA 230/100 kV transformers with an AC side filtering capability of 40 MVAR. The DC side filters only consisted of a third harmonic filter with the smoothing reactor. Under normal operating conditions, VSC-1 and VSC-3 were operated in voltage-control mode, while VSC-2 and VSC-4 were engaged in power-control mode. Eight relaying units were placed at each end of the lines. The purpose was to extract current and voltage waveforms from both positive and negative poles and establish protection decisions based on their features.

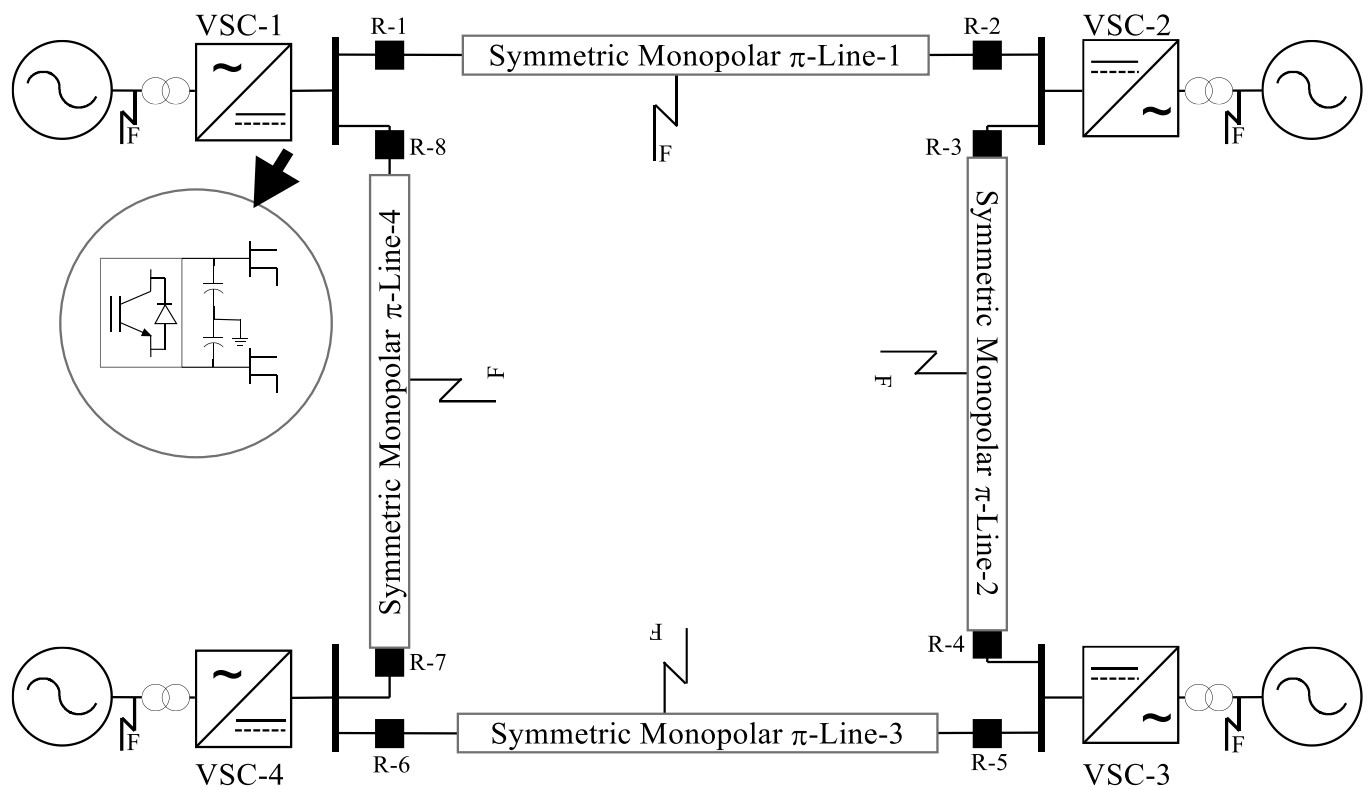

**Figure 1.** Studied symmetrical monopolar VSC-MT-HVDC system.

To emulate the accurate behavior of the physical system, the model was simulated at a very high sampling frequency of 135 kHz. However, considering industrial standards and available hardware for data acquisition, the voltage and current waveforms were acquired at a far lower sampling frequency of 3200 Hz or 64 samples per cycle of the AC side voltage. The detailed parameters of the AC grid and converter stations are presented in Table 1.

**Table 1.** AC System and Converter parameters.

| System | Parameter |
|---|---|
| AC Systems | 230 kV, 2000 MVA, 50 Hz |
| Transformers | Yg-D configuration, 230/100 kV, 250 MVA, $X_L = 0.15\ pu$ |
| Phase Reactor | 0.15 pu |
| High-Pass AC Filter | Tuned: 27th Harmonic, 18 MVAr, Q = 15 |
| | Tuned: 54th Harmonic, 22 MVAr, Q = 15 |
| 3rd Harmonic DC Filter | $R = 0.14737\ \blacksquare,\ L = 46.908\ mH,\ C = 12\ \mu F$ |
| Smoothing Inductor | $R = 0.0251\ \blacksquare,\ L = 8\ mH$ |
| Converter | 100 MVA, rated DC voltage = 200 kV ($\pm100\ kV$), Rated DC current = 500 A |
| Transmission Lines | Symmetric monopolar lines, PI-configuration, Length = 100 km, R = 0.0139 Ω/Km, L = 0.159 mH/Km, C = 0.231 μF/Km |

## 3. Methodology

The objective of this work is divided into two folds: (i) effective fault types classification (internal (DC)/external (AC or DC)) and (ii) accurate prediction of fault event locations. Figure 2 shows the flowchart of the presented work that has been adopted for fault classification and location in the HVDC transmission system. The proposed classification scheme

was designed based on the positive and negative pole voltage and current waveforms obtained at the relaying points (R-1 to R-8) present at each end of the transmission line. Afterward, different statistical features such as mean, standard deviation, entropy, and correlation coefficient of voltage and current were extracted in order to train and test the data-mining models. In the DM framework, five state-of-the-art ML models (ANN, SVM, KNN, RF, and ELM) and one DL model (DBN) were used for training and testing in order to recognize the fault types. In order to increase the efficiency of the DM model by reducing the computational burden, the sequential forward feature selection (SFS) was integrated with each ML-based DM model. Here, the accuracy of fault event classification was analyzed in a comparative manner. Again with an intention to predict the accurate fault location, six state-of-the-art ML-based regressors (ANN, SVR, KNN, ELM, LR, and RF) and one DL-based regressor (DBN) were used for training and testing. In this case, several error indices such as MSE, MAPE, RMSE, NRMSE, and R-value were calculated and compared for each model. For classification (fault type recognition) and regression (prediction of fault location) analysis, 5-fold cross-validation was used to validate the performance of each model.

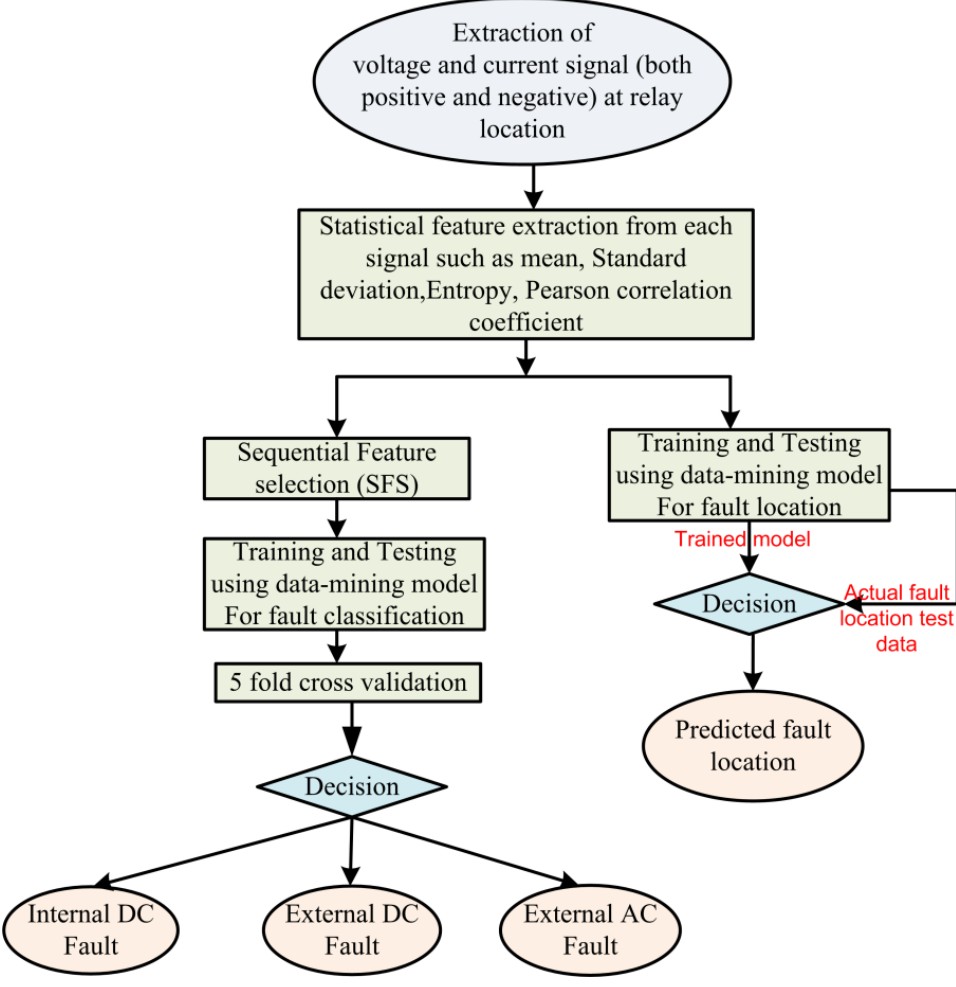

**Figure 2.** Flowchart of the proposed methodology of HVDC fault classification and location.

### 3.1. Fault and No-Fault Data Generation

For training the data-mining models, a data window of 32 samples was taken 0.01 s after the inception of the faults.

Fault case simulation includes:

1.  Four types of faults such as positive pole-to-ground (P-G), negative pole-to-ground (N-G), pole-to-pole (P-P), and pole-to-pole-to-ground (P-P-G) were simulated in all 4 lines (Line-1 through Line-4).
2.  Thirteen (13) fault resistances, namely $R_{fault}$ = 0.01 Ω, 0.1 Ω, 1 Ω, 10 Ω, 20 Ω, 30 Ω, 40 Ω, 50 Ω, 60 Ω, 70 Ω, 80 Ω, 90 Ω, and 100 Ω were considered during the simulation of fault cases.
3.  Nine (9) fault locations in each line were considered during the fault simulation, namely 10%, 20%, 30%, 40%, 50%, 60%, 70%, 80%, and 90% of the line.

Therefore, a total of 4 × 4 × 13 × 9 = 1872 fault cases were simulated. Each fault case in a line when considered from the relays present at both ends of the line was equivalent to 2 cases. The total number of fault cases presented to the classifier was 1872 × 2 = 3744.

No-fault transient cases presented to the classifier were as follows:

4.  External DC fault cases: These were the voltage and current waveforms at the relaying points of a line during a DC fault in another line. For example, during a fault in Line-1, voltage and current waveforms obtained at the relaying points R-3, R-4, R-5, R-6, R-7, and R-8 were taken as external DC fault cases. The total number of external DC fault cases present was 11,232.
5.  External AC fault cases: external AC faults were simulated at each of 4 AC sides as follows
    - Five types of faults A-G, A-B, A-B-G, A-B-C, and A-B-C-G were simulated in all 4 AC sides.
    - Thirteen (13) fault resistances, $R_{fault}$ = 0.01 Ω, 0.1 Ω, 1 Ω, 10 Ω, 20 Ω, 30 Ω, 40 Ω, 50 Ω, 60 Ω, 70 Ω, 80 Ω, 90 Ω, and 100 Ω, were considered during the simulation of fault cases.
    - The number of external AC fault cases present was 2080. Therefore, the total no-fault transient cases present for classification was 13,312.

### 3.2. Feature Extraction

After the extraction of the desired voltage and current signal at the relay's location, the following fourteen statistical features (F1 to F14) were extracted and used by the ML or data-mining model to automatic recognition of HVDC faults type and location:

F1: mean of positive pole voltage;
F2: mean of negative pole voltage;
F3: mean of positive pole current;
F4: mean of negative pole current;
F5: standard deviation of positive pole voltage;
F6: standard deviation of negative pole voltage;
F7: standard deviation of positive pole current;
F8: standard deviation of negative pole current;
F9: entropy of positive pole voltage;
F10: entropy of negative pole voltage;
F11: entropy of positive pole current;
F12: entropy of negative pole current;
F13: Pearson correlation coefficient between positive pole voltage and current;
F14: Pearson correlation coefficient between negative pole voltage and current.

A sample of the features' magnitude plot with respect to time is shown visually in Figure 3 (a positive line-to-ground fault at 10% of the transmission line with fault resistance of 0.02 ohm). Moreover, the variation of features with respect to fault resistance ($R_f$), fault location ($L_f$), and fault types are also presented in tabular form in Tables 2–4, respectively. Table 2 depicts the features at R1 for a positive pole-to-ground fault at 50% of Line-1 with the variation of fault resistance ($R_f$). Features at R1 for a positive pole-to-ground fault at Line-1 with a particular fault resistance ($R_f$ = 1 ohm) are tabulated in Table 3 with the variation in fault locations. Similarly, features at R1 for a fault at 50% of Line-1 with fault resistance $R_f$

equal to 1 ohm are tabulated in Table 4 for different types of DC faults. As shown in the figures and tables, all the features were found to be sensitive to fault occurrences and can be useful for the fault recognition process.

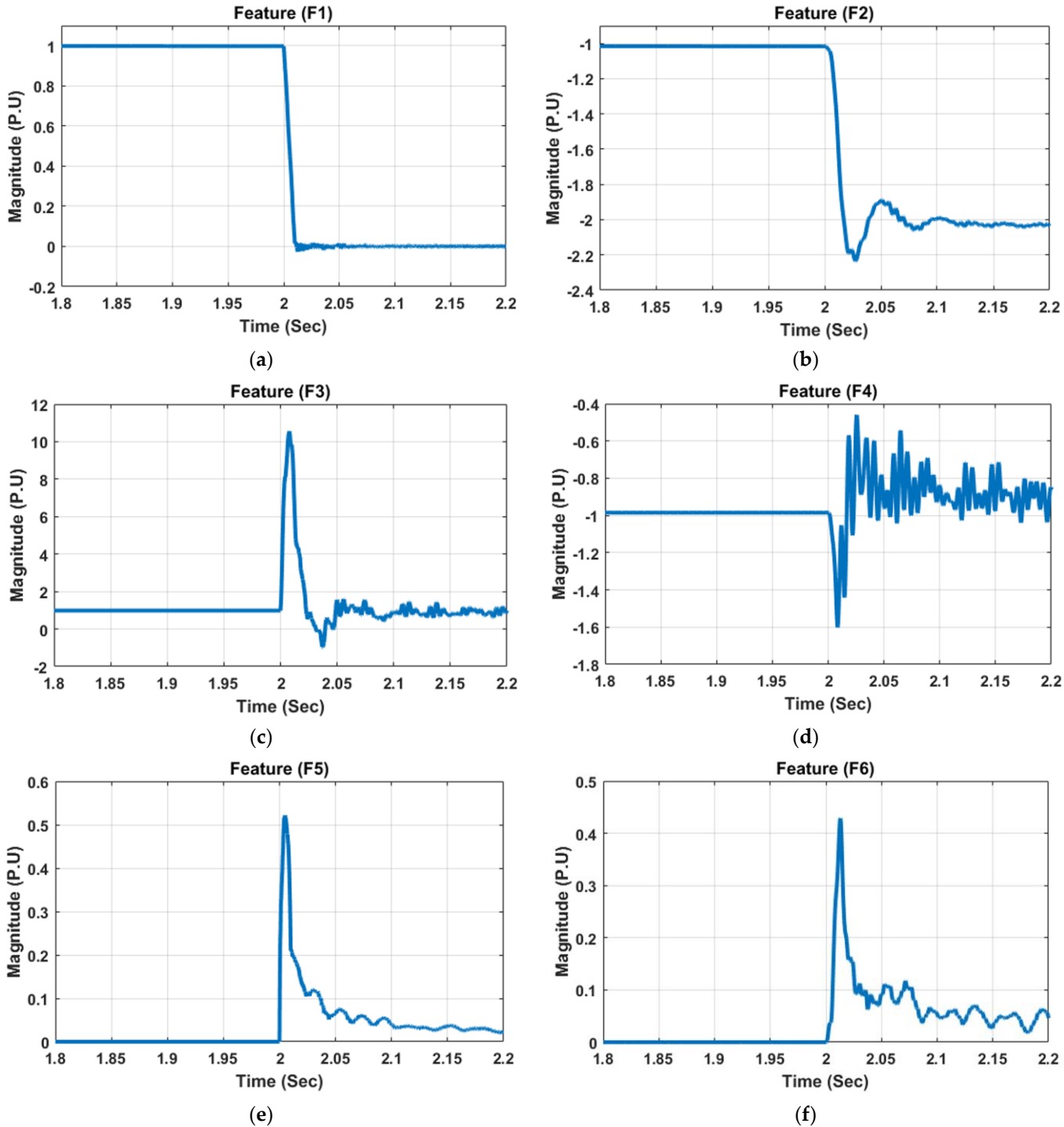

**Figure 3.** *Cont.*

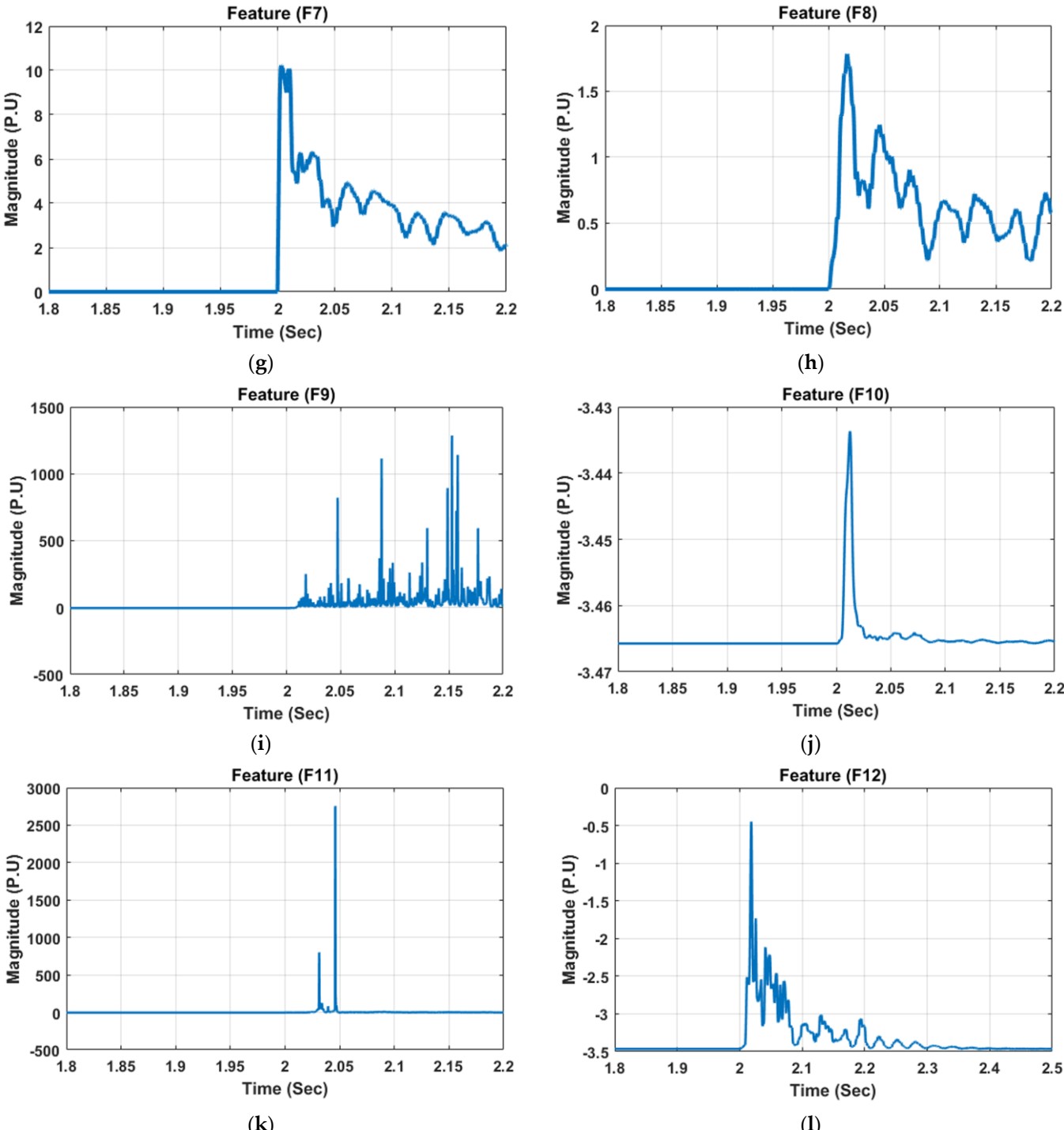

**Figure 3.** *Cont.*

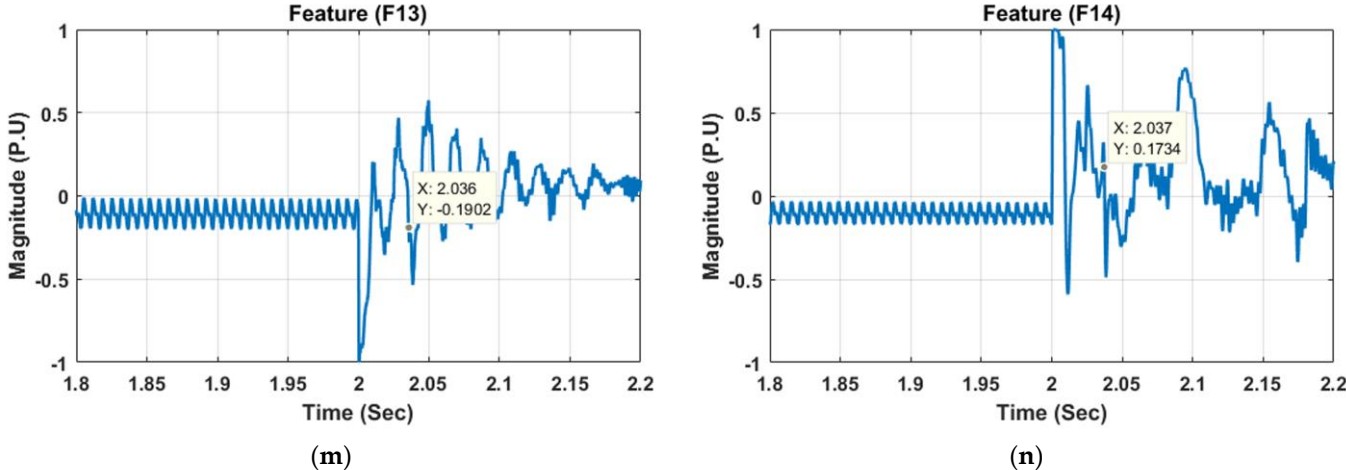

**Figure 3.** Sample features for a positive line-to-ground fault at 10% of the transmission line with fault resistance of 0.02 ohm. (**a**) Feature F1, (**b**) Feature F2, (**c**) Feature F3, (**d**) Feature F4, (**e**) Feature F5, (**f**) Feature F6, (**g**) Feature F7, (**h**) Feature F8, (**i**) Feature F9, (**j**) Feature F10, (**k**) Feature F11, (**l**) Feature F12, (**m**) Feature F13, (**n**) Feature F14.

**Table 2.** Features at R1 for a positive pole-to-ground fault at 50% of Line-1.

| $R_f$ | F1 | F2 | F3 | F4 | F5 | F6 | F7 | F8 | F9 | F10 | F11 | F12 | F13 | F14 |
|---|---|---|---|---|---|---|---|---|---|---|---|---|---|---|
| 0.01 | 0.05 | −1.33 | 8.10 | −1.28 | 0.34 | 0.33 | 7.82 | 0.22 | 2.99 | −3.44 | −2.89 | −3.45 | −0.03 | 0.07 |
| 0.10 | 0.06 | −1.32 | 8.08 | −1.28 | 0.33 | 0.32 | 7.64 | 0.22 | 2.07 | −3.44 | −2.93 | −3.45 | −0.01 | 0.08 |
| 1.00 | 0.13 | −1.28 | 7.82 | −1.27 | 0.30 | 0.28 | 6.13 | 0.21 | −1.41 | −3.44 | −3.14 | −3.45 | 0.11 | 0.14 |
| 10.00 | 0.50 | −1.12 | 5.30 | −1.14 | 0.21 | 0.11 | 1.83 | 0.13 | −3.39 | −3.46 | −3.40 | −3.46 | 0.29 | 0.11 |
| 20.00 | 0.66 | −1.08 | 3.97 | −1.08 | 0.15 | 0.06 | 1.01 | 0.08 | −3.44 | −3.46 | −3.43 | −3.46 | 0.16 | 0.24 |
| 30.00 | 0.75 | −1.06 | 3.25 | −1.05 | 0.12 | 0.04 | 0.73 | 0.06 | −3.45 | −3.46 | −3.44 | −3.46 | 0.06 | 0.21 |
| 40.00 | 0.80 | −1.05 | 2.81 | −1.04 | 0.10 | 0.03 | 0.58 | 0.05 | −3.46 | −3.47 | −3.44 | −3.46 | 0.00 | 0.18 |
| 50.00 | 0.83 | −1.04 | 2.51 | −1.03 | 0.08 | 0.03 | 0.49 | 0.04 | −3.46 | −3.47 | −3.44 | −3.46 | −0.05 | 0.21 |
| 60.00 | 0.85 | −1.04 | 2.30 | −1.02 | 0.07 | 0.02 | 0.43 | 0.04 | −3.46 | −3.47 | −3.45 | −3.47 | −0.08 | 0.21 |
| 70.00 | 0.87 | −1.04 | 2.14 | −1.02 | 0.07 | 0.02 | 0.39 | 0.03 | −3.46 | −3.47 | −3.45 | −3.47 | −0.10 | 0.22 |
| 80.00 | 0.89 | −1.03 | 2.01 | −1.01 | 0.06 | 0.02 | 0.35 | 0.03 | −3.46 | −3.47 | −3.45 | −3.47 | −0.12 | 0.22 |
| 90.00 | 0.90 | −1.03 | 1.91 | −1.01 | 0.05 | 0.02 | 0.32 | 0.03 | −3.46 | −3.47 | −3.45 | −3.47 | −0.13 | 0.23 |
| 100.00 | 0.91 | −1.03 | 1.83 | −1.01 | 0.05 | 0.01 | 0.30 | 0.02 | −3.46 | −3.47 | −3.45 | −3.47 | −0.14 | 0.23 |

**Table 3.** Features at R1 for a positive line-to-ground fault at Line-1 with $R_f$ = 1 ohm.

| $L_f$ | F1 | F2 | F3 | F4 | F5 | F6 | F7 | F8 | F9 | F10 | F11 | F12 | F13 | F14 |
|---|---|---|---|---|---|---|---|---|---|---|---|---|---|---|
| 10% | 0.11 | −1.28 | 9.43 | −1.40 | 0.23 | 0.25 | 7.93 | 0.72 | −1.81 | −3.45 | −3.04 | −3.27 | 0.26 | −0.05 |
| 20% | 0.11 | −1.28 | 8.94 | −1.39 | 0.24 | 0.26 | 6.85 | 0.52 | −1.55 | −3.45 | −3.14 | −3.38 | 0.18 | −0.01 |
| 30% | 0.11 | −1.28 | 8.64 | −1.36 | 0.27 | 0.27 | 6.58 | 0.36 | −1.37 | −3.45 | −3.16 | −3.43 | 0.11 | 0.02 |
| 40% | 0.12 | −1.28 | 8.24 | −1.32 | 0.29 | 0.27 | 6.32 | 0.25 | −1.41 | −3.44 | −3.16 | −3.45 | 0.11 | 0.08 |
| 50% | 0.13 | −1.28 | 7.82 | −1.27 | 0.30 | 0.28 | 6.13 | 0.21 | −1.41 | −3.44 | −3.14 | −3.45 | 0.11 | 0.14 |
| 60% | 0.13 | −1.28 | 7.39 | −1.23 | 0.32 | 0.28 | 6.01 | 0.26 | −1.19 | −3.44 | −3.09 | −3.44 | 0.12 | 0.18 |
| 70% | 0.13 | −1.29 | 6.95 | −1.19 | 0.33 | 0.28 | 5.88 | 0.39 | −1.19 | −3.44 | −3.05 | −3.42 | 0.12 | 0.20 |
| 80% | 0.14 | −1.29 | 6.54 | −1.16 | 0.36 | 0.29 | 5.70 | 0.56 | −0.98 | −3.44 | −3.00 | −3.36 | 0.11 | 0.23 |
| 90% | 0.14 | −1.29 | 6.15 | −1.15 | 0.37 | 0.29 | 5.49 | 0.79 | −0.81 | −3.44 | −2.97 | −3.25 | 0.10 | 0.28 |

**Table 4.** Features at R1 for a fault at 50% of Line-1 with $R_f$ = 1 ohm.

| $F_T$ | F1 | F2 | F3 | F4 | F5 | F6 | F7 | F8 | F9 | F10 | F11 | F12 | F13 | F14 |
|------|------|-------|-------|--------|------|------|------|------|-------|-------|-------|-------|-------|-------|
| P−G | 0.13 | −1.28 | 7.82 | −1.27 | 0.30 | 0.28 | 6.13 | 0.21 | −1.41 | −3.44 | −3.14 | −3.45 | 0.11 | 0.14 |
| N−G | 1.28 | −0.13 | 1.27 | −7.82 | 0.28 | 0.30 | 0.21 | 6.13 | −3.44 | −1.42 | −3.45 | −3.14 | 0.12 | 0.11 |
| P−P | 0.20 | −0.20 | 13.28 | −13.28 | 0.32 | 0.32 | 5.44 | 5.44 | −2.33 | −2.33 | −3.37 | −3.37 | −0.29 | −0.29 |
| PP−G | 0.27 | −0.27 | 12.71 | −12.71 | 0.28 | 0.28 | 4.92 | 4.92 | −2.82 | −2.82 | −3.38 | −3.38 | −0.22 | −0.22 |

## 4. Studied DMTs for HVDC Fault Recognition

After extraction of the features vector, all the features were inputted to the DMTs in order to recognize the faults in the HVDC system. In the framework of data mining, the following techniques (as mentioned in Sections 4.1–4.8) were implemented in this work individually in order to gauge the suitability of their implementation in the HVDC system. It is well known that in the DMTs, feature selection is one of the most important processes as it helps to increase the prediction accuracy by choosing the most sensitive attributes and rejecting the redundant and irrelevant ones. In this regard, the SFS method was used in this work with different DMTs for the selection of the best feature vector.

### 4.1. Artificial Neural Network (ANN)

ANN classifiers were adopted for various disturbance detections and classifications in the power system because of their simplicity, flexibility, and quick response time. ANN is one of the most applied DMTs in the HVDC system. The ANN concept was inspired by the biological nerve cells. Generally, it consists of three fundamental layers such as input, hidden, and output layers [56].

### 4.2. Support Vector Machine (SVM)

SVM is a very popular supervised learning classification approach widely being used for fault events recognition in power systems. It can be used for the analysis of data by both classifications as well as regression problems [57]. With SVM, a novel training method was introduced that helped to optimize the boundary amongst distinct classes. The main aim of this classifier was to create a decision boundary. The best-classified decision boundary is known as a hyperplane. This algorithm consists of the following data sets: input feature vector and class levels $(A_j, B_j)$, where $j = 1, 2, \ldots \ldots n$, and $n$ represents the number of classes. The result of SVM (penalty factor) is derived by combining Equations (1) and (2):

$$P(m, \xi) = 0.5\left(m^T m\right) + C\sum_{j=1}^{1} \xi_j : \xi_j > 0 \tag{1}$$

and

$$B_j\left(m^T \varphi\left(A_j\right) + b\right) \geq 1 - \xi_j \tag{2}$$

where $m$ represents weights, $b$ represents bias, $\xi$ represents error, and $P$ represents penalty factor. This approach is highly productive in higher dimensional space and provides better accuracy. Different kernel functions (such as linear, radial, polynomial, and sigmoid) can be used for mapping the input data to increase the accuracy [56].

### 4.3. Support Vector Regression (SVR)

SVR is one of the popularly used ML algorithms based on supervised learning methods and is concerned with the classification resulting in the prediction of discrete values. This algorithm is being implemented on various power systems and renewable system modeling applications with extremely promising results. Both SVR and SVM share the same fundamental concept, but the former helps in the prediction of real values [57]. The added advantage of this ML algorithm is that it works perfectly on non-linear data, which makes

it superior to linear regression. It works on three parameters such as kernel, hyperplane, and decision boundary.

The distance between the decision boundary and hyperplane is an epsilon value $c$. Therefore, both side distances are considered as $+c$ and $-c$. The equations of the decision boundary are given as

$$\begin{aligned} ux + d &= +c \\ ux + d &= -c \end{aligned} \tag{3}$$

Therefore, the equation of the hyperplane is written as: $z = ux + d$. Thus, the hyperplane which satisfies SVR criteria lies between: $-c < z - ux + d < +c$. The computational complexity independent of the dimension of the input space is the major advantage of this technique.

### 4.4. Extreme Learning Machine (ELM)

ELM is a widely used single hidden layer feed-forward neural network. ELM is considered as a fast and robust training algorithm and can be useful in fault detection in power systems with reduced computational complexity. On a wide scope, it performs several real-time learning tasks including pattern classification, clustering, and regression based on the type of datasets. ELM works on two basic steps such as random initialization and linear parameter solution. There are several hidden neurons in the ELM, and their input weights are allocated arbitrarily [58]. The mathematical expression for the ELM output function with n number of hidden nodes is represented as:

$$f_n(x) = \sum_{i=1}^{n} \theta_i b_i(x) \tag{4}$$

where $\theta_i$ is the weight of the $i^{th}$ hidden node and $b(x) = [b_i(x) \ldots \ldots b_n(x)]$ represents output mapping.

$b_i(x) = G(u_i, v_i, x)$, where $u_i$ and $v_i$ represent the parameters of the $i^{th}$ hidden node.

### 4.5. K-Nearest Neighbour (KNN)

KNN is a simple and non-parametric algorithm in ML based on a supervised learning approach implemented for faults in the HVDC transmission system. It first stores the newly assigned datasets and then classifies them accordingly (lazy learning method). This algorithm then compares the new datasets with the existing datasets. It calculates the Euclidean distance between the data points. The basic steps of KNN are (i) calculate the distance, (ii) find the closest neighbors, (iii) vote on the level, and (iv) take the majority vote [59]. The mathematical expression for distance calculation is as follows:

$s = $ the eigenvectors are represented as $<p_1(s), p_2(s), \ldots \ldots p_n(s)>$

For an instance 's', the eigenvector is represented as $[p_1(s), p_2(s), \ldots \ldots p_n(s)]$ where $p_1(s)$ represents the $n^{th}$ attribute value of the instance $s$.

The distance between two instances $s_i$ and $s_j$ is defined as $D(s_i, s_j)$, where

$$D(s_i, s_j) = \sqrt{\sum_{b=1}^{n} (p_b(s_i) - p_b(s_j))^2} \tag{5}$$

As no training is initially required, the addition of new data is possible and it does not affect the accuracy of this ML algorithm. It is simple to put into practice and resistant to noisy training data.

### 4.6. Random Forest Algorithm (RF)

RF is one of the widely used ML algorithms based on supervised learning methods and is used for both the classification and regression of datasets. It assembles a wide range of classifiers for solving complex datasets. The classifiers are constituted with different decision trees (DTs). Each individual tree in the random forest spit out a class prediction,

and the class with the most votes became our model's prediction. More DTs tend to have more accuracy and, at the same instance, reduce overfitting. In the beginning, N numbers of decision trees were combined to generate the RF, and then predictions were made for each tree that was produced. It is one of the less time-consuming algorithms and enhances a wider range of accuracy [60].

### 4.7. Linear Regression Algorithm (LR)

LR is one of the simplest and most often used DMT used for regression. It is a statistical technique for performing predictive analysis on wider sets of data. This algorithm shows a linear relationship between predicted and target values. The major function of linear regression is to find the best fit. The difference between the predicted and targeted value is called the error. This error should be lowered to find the best fit. This algorithm is easy to implement because of its linearity and it handles overfitting as well as cross-validation.

Linear regression can be represented mathematically as:

$$T = \alpha_\theta + \alpha_1 q + e \tag{6}$$

where $T$ = dependent variable or targeted variable, $q$ = independent variable or predictor variable, $\alpha_\theta$ = intercept of the line, $\alpha_1$ = linear regression coefficient, and $e$ = random error.

### 4.8. Deep Belief Network (DBN)

The concept of DBN was developed by Geoffrey Hinton in 2006. DBN is an advanced hybrid generative model of unsupervised deep learning technique, intended to address the lacunas with conventional neural networks. DBNs are a more computationally dynamic and robust type of feedforward neural network that may be utilized for various fault detections in HVDC transmission systems. This algorithm consists of combining various layers of stochastic latent variables. These stochastic variables consist of both binary latent variables as well as hidden units. These are referred to as stochastic variables since they have a chance of taking on any value within a given range. DBNs have no direction in the first two layers. After that, this network has direct linkages to other layers present inside it. Due to their capability to function as generating and discriminative models, DBNs vary from conventional neural networks. This network forms hierarchical layers forming both top (associative memory) and bottom (visible unit) layers [60,61].

It comprises many completely linked layers of restricted Boltzmann machines (RBM). This RBM comprises hidden layers. RBM serves as a probability distribution model associated with energy. Suppose the hidden layers are represented as $p$ and the visible layer is represented as $q$, respectively. The energy function can be expressed as:

$$E(p, q|\phi) = -\sum_{n=1}^{u} \alpha_n p_n - \sum_{m=1}^{x} \beta_m q_m - \sum_{n=1}^{u} \sum_{m=1}^{x} p_n \omega_{mn} q_m \tag{7}$$

where $p_n$ represents the state values of visible layers,

$q_m$ represents the state values of hidden layers,

$\omega$ represents the weight matrix, $\alpha$ represents the bias vectors for the visible layer, and $\beta$ represents the bias vectors for the hidden layer.

The joint probability distribution function can be given as $D(p, q|\phi) = \frac{e^{-E(p,q|\phi)}}{Y(\phi)}$.

## 5. Results and Discussion

In this section, a complete fault assessment of the considered HVDC system using different DMTs has been comprehensively carried out in a MATLAB/SIMULINK environment. As per the flowchart presented in Figure 2, a total of fourteen statistical features from the voltage and current signal (both positive and negative pole) at relying points were extracted and fed as input to each DMT. As presented in Figure 2, the objective of the work

was divided into two groups, namely (i) fault events classification and (ii) fault location. The results and assessment of each objective are mentioned in the subsequent subsections.

*5.1. Fault Events Classification*

As presented in Section 3.1, the faults simulated in the considered HVDC systems were classified into three important classes: Class-1 (internal DC faults), Class-2 (external DC faults), and Class-3 (external AC faults). For this instance, if a fault occurred in Line-1, voltage and current waveforms obtained at the relaying points R-3, R-4, R-5, R-6, R-7, and R-8 would be taken as external DC fault cases, whereas the signal measured at R-1 and R-2 would give rise to an internal DC fault. In this regard, 3744 numbers of fault samples were simulated for Class-1, whereas 11,232 and 2088 numbers of samples were simulated for Class-2 and Class-3, respectively.

The above-mentioned faulty signal samples were then classified using different DMTs as mentioned in Section 4 according to the extracted features. The performance of the DMTs was analyzed through the output confusion matrix and in terms of the accuracy of the classifier. The mathematical formulation of accuracy is mentioned in Figure 4.

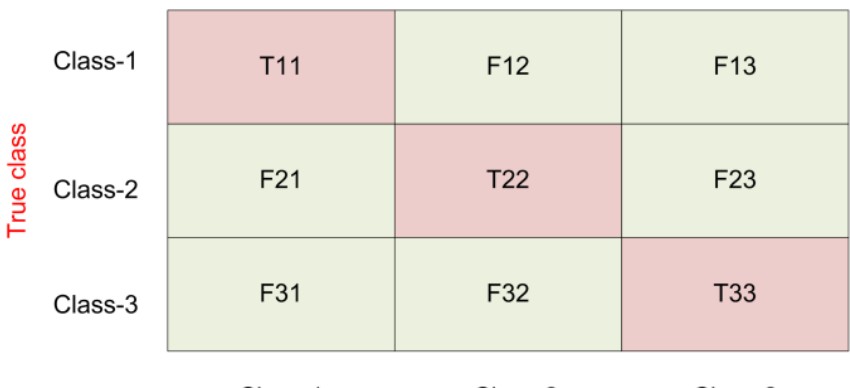

$$\text{Accuracy} = \frac{(T11+T22+T33)}{(T11+F12+F13+F21+T22+F23+F31+F32+T33)}*100\%$$

**Figure 4.** Pictorial representation of confusion matrix and mathematical formulation of accuracy.

It is well-known that in the DMTs, feature selection is one of the most important processes as it helps to increase the prediction accuracy by choosing the most sensitive attributes and rejecting the redundant and irrelevant ones. Here, the sequential forward selection (SFS) was used in this work with different DMTs for the selection of the best feature vector. Initially, the ML techniques ANN, ELM, SVM, RF, and KNN were trained and tested with five-fold cross-validation. The output results are presented in Tables 5–9, respectively. The confusion matrix corresponding to each ML technique is presented in Figures 5–9, respectively. As shown in Table 5 and Figure 5, the maximum accuracy was found to be 91.5% with 12 numbers of features utilized by the ANN classifier. Here, it can be seen that, out of 3744 internal DC faults events, only 2538 events were truly detected as Class-1 (otherwise called dependability of fault detection) which was 67.8%. The formula for dependability is presented in Equation (8). However, the AC faults were recognized with 95% detection accuracy, whereas the external DC faults were detected with 97.9% accuracy or true-positive rate (TPR). From this analysis, it can be analyzed that although the overall percentage accuracy in the case of ANN was found to be 91.5%, more than 32.2% of internal DC faults were incorrectly detected as external DC faults which occurred

the far end of the transmission line. Compared with ANN, the performance of the ELM classifier was found to be inferior in terms of both the TPR and false negative rate (FNR) of each fault class. With 12 numbers of features, the ELM classifier provides a maximum of 84% overall classification accuracy, where the value of dependability (TPR of Class 1) was found to be 62% (refer to Figure 6 and Table 6).

$$\text{Dependibility} = \frac{\text{Numbers of internal DC faults events truely detected as Class-1}}{\text{Total numbers of internal DC faults events tested}} * 100\% \tag{8}$$

**Table 5.** Performance of fault detection using SFS with ANN (five-fold cross-validation).

| No. of Features | Selected Features | Accuracy (%) |
|---|---|---|
| 1 | F11 | 73.52 |
| 2 | F11,F14 | 80.32 |
| 3 | F11,F14,F3 | 82.29 |
| 4 | F11,F14,F3,F7 | 85.38 |
| 5 | F11,F14,F3,F7,F8 | 86.39 |
| 6 | F11,F14,F3,F7,F8,F5 | 88.68 |
| 7 | F11,F14,F3,F7,F8,F5,F12 | 88.23 |
| 8 | F11,F14,F3,F7,F8,F5,F12,F10 | 89.90 |
| 9 | F11,F14,F3,F7,F8,F5,F12,F10,F6 | 89.85 |
| 10 | F11,F14,F3,F7,F8,F5,F12,F10,F6,F13 | 90.08 |
| 11 | F11,F14,F3,F7,F8,F5,F12,F10,F6,F13,F9 | 90.20 |
| **12** | **F11,F14,F3,F7,F8,F5,F12,F10,F6,F13,F9,F1** | **91.50** |
| 13 | F11,F14,F3,F7,F8,F5,F12,F10,F6,F13,F9,F1,F4 | 91.42 |
| 14 | F11,F14,F3,F7,F8,F5,F12,F10,F6,F13,F9,F1,F4,F2 | 89.68 |

**Table 6.** Performance of fault detection using SFS with ELM (five-fold cross-validation).

| No. of Features | Selected Features | Accuracy (%) |
|---|---|---|
| 1 | F3 | 65.90 |
| 2 | F3,F8 | 79.28 |
| 3 | F3,F8,F11 | 79.43 |
| 4 | F3,F8,F11,F13 | 79.54 |
| 5 | F3,F8,F11,F13,F6 | 82.00 |
| 6 | F3,F8,F11,F13,F6,F10 | 82.08 |
| 7 | F3,F8,F11,F13,F6,F10,F2 | 83.24 |
| 8 | F3,F8,F11,F13,F6,F10,F2,F14 | 83.58 |
| 9 | F3,F8,F11,F13,F6,F10,F2,F14,F5 | 83.51 |
| 10 | F3,F8,F11,F13,F6,F10,F2,F14,F5,F9 | 83.47 |
| 11 | F3,F8,F11,F13,F6,F10,F2,F14,F5,F9,F12 | 83.59 |
| **12** | **F3,F8,F11,F13,F6,F10,F2,F14,F5,F9,F12,F1** | **84.00** |
| 13 | F3,F8,F11,F13,F6,F10,F2,F14,F5,F9,F12,F1,F7 | 83.60 |
| 14 | F3,F8,F11,F13,F6,F10,F2,F14,F5,F9,F12,F1,F7,F4 | 83.57 |

**Table 7.** Performance of fault detection using SFS with SVM (five-fold cross-validation).

| No. of Features | Selected Features | Accuracy (%) |
|---|---|---|
| 1 | F3 | 68.7 |
| 2 | F3,F2 | 70.1 |
| 3 | F3,F2,F8 | 73.7 |
| 4 | F3,F2,F8,F12 | 75.6 |
| 5 | F3,F2,F8,F12,F6 | 78.7 |
| 6 | F3,F2,F8,F12,F6,F1 | 83.6 |
| 7 | F3,F2,F8,F12,F6,F1,F10 | 83.9 |
| 8 | F3,F2,F8,F12,F6,F1,F10,F5 | 84.7 |
| 9 | F3,F2,F8,F12,F6,F1,F10,F5,F11 | 87.8 |
| 10 | F3,F2,F8,F12,F6,F1,F10,F5,F11,F4 | 89.9 |
| 11 | F3,F2,F8,F12,F6,F1,F10,F5,F11,F4,F7 | 89.9 |
| 12 | F3,F2,F8,F12,F6,F1,F10,F5,F11,F4,F7,F13 | 96.9 |
| 13 | F3,F2,F8,F12,F6,F1,F10,F5,F11,F4,F7,F13,F14 | 97.1 |
| **14** | **F3,F2,F8,F12,F6,F1,F10,F5,F11,F4,F7,F13,F14,F9** | **97.3** |

**Table 8.** Performance of fault detection using SFS with RF (five-fold cross-validation).

| No. of Features | Selected Features | Accuracy (%) |
|---|---|---|
| 1 | F4 | 68.8 |
| 2 | F4,F3 | 68.1 |
| 3 | F4,F3,F1 | 71.4 |
| 4 | F4,F3,F1,F2 | 78.7 |
| 5 | F4,F3,F1,F2,F7 | 78.7 |
| 6 | F4,F3,F1,F2,F7,F5 | 81.6 |
| 7 | F4,F3,F1,F2,F7,F5,F6 | 82.4 |
| 8 | F4,F3,F1,F2,F7,F5,F6,F8 | 83.0 |
| 9 | F4,F3,F1,F2,F7,F5,F6,F8,F9 | 89.0 |
| 10 | F4,F3,F1,F2,F7,F5,F6,F8,F9,F10 | 88.9 |
| 11 | F4,F3,F1,F2,F7,F5,F6,F8,F9,F10,F11 | 90.4 |
| 12 | F4,F3,F1,F2,F7,F5,F6,F8,F9,F10,F11,F12 | 93.7 |
| 13 | F4,F3,F1,F2,F7,F5,F6,F8,F9,F10,F11,F12,F13 | 96.4 |
| **14** | **F4,F3,F1,F2,F7,F5,F6,F8,F9,F10,F11,F12,F13,F14** | **96.7** |

A similar kind of observation can be seen for SVM and RF but with a little improvement in the performance (accuracy and dependability), such as 97.3% and 89.9% and 96.7% and 86.9%, respectively (refer to Tables 7 and 8 and Figures 7 and 8). However, the performance was significantly improved in the case of the KNN classifier. In the case of KNN (refer to Figure 9), out of 3744 internal DC faults events only 3599 numbers of events were truly detected as Class-1 having a dependability of 96.1%. However, the AC faults were recognized with 100% detection accuracy, whereas the external DC faults were detected with 95.5% accuracy. From this analysis, it can be analyzed that the overall maximum percentage accuracy in the case of KNN was found to be 98.82% (refer to Table 9), where only 4.1% of internal DC faults were incorrectly detected as external faults and 0.5% of other external DC faults were incorrectly detected as internal faults.

**Table 9.** Performance of fault detection using SFS with KNN (five-fold cross-validation).

| No. of Features | Selected Features | Accuracy (%) |
|---|---|---|
| 1 | F3 | 64.72 |
| 2 | F3,F12 | 89.33 |
| 3 | F3,F12,F6 | 94.45 |
| 4 | F3,F12,F6,F4 | 97.72 |
| 5 | F3,F12,F6,F4,F5 | 97.83 |
| 6 | F3,F12,F6,F4,F5,F10 | 97.8 |
| 7 | F3,F12,F6,F4,F5,F10,F13 | 98.42 |
| 8 | F3,F12,F6,F4,F5,F10,F13,F14 | 98.62 |
| 9 | F3,F12,F6,F4,F5,F10,F13,F14,F1 | 98.73 |
| **10** | **F3,F12,F6,F4,F5,F10,F13,F14,F1,F2** | **98.82** |
| 11 | F3,F12,F6,F4,F5,F10,F13,F14,F1,F2,F7 | 98.81 |
| 12 | F3,F12,F6,F4,F5,F10,F133,F14,F1,F2,F7,F8 | 98.78 |
| 13 | F3,F12,F6,F4,F5,F10,F13,F14,F1,F2,F7,F8,F9 | 98.74 |
| 14 | F3,F12,F6,F4,F5,F10,F13,F14,F1,F2,F7,F8,F9,F11 | 98.75 |

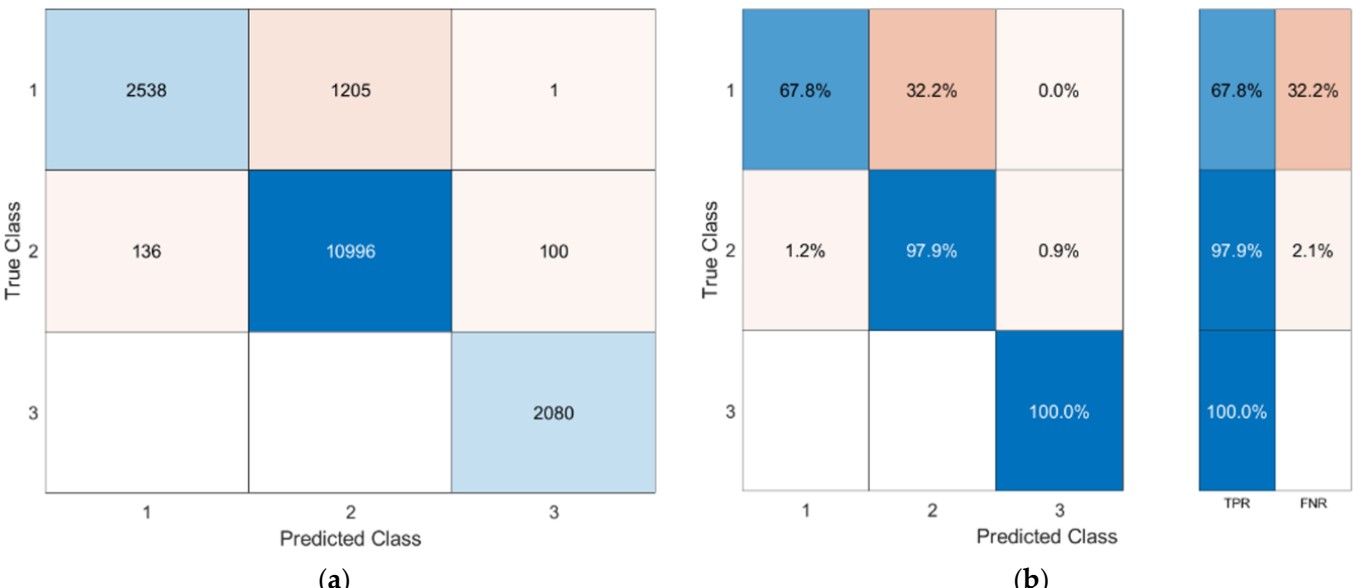

(**a**)　　　　　　　　　　　　　　　(**b**)

**Figure 5.** Confusion matrix corresponding to the highest accuracy as shown in Table 5: (**a**) confusion matrix showing numerical values (**b**) TPR and FNR (in percentages).

In the studied data-mining framework, the authors tested the extracted fault dataset with a deep learning technique named 'DBM'. The general architecture and theory of DBN are presented in Section 4.8. In this case, we used all the features for the training of DBN. Similar five-fold cross-validation was performed to analyze the output results. The output confusion matrix corresponding to the DBN classifier is shown in Figure 10. As shown in Figure 10, the overall accuracy of the classifier was found to be 98.9% with 96.1% dependability or TPR for Class-1. Although the classifier performed better compared with other ML-based DMTs to classify the fault, the average testing time was slightly higher (refer to Table 10). Moreover, Table 10 depicts the performance of each DMT in noisy conditions. Here, we have noted a similar kind of observation as presented for without noisy data, but with a reduced accuracy level.

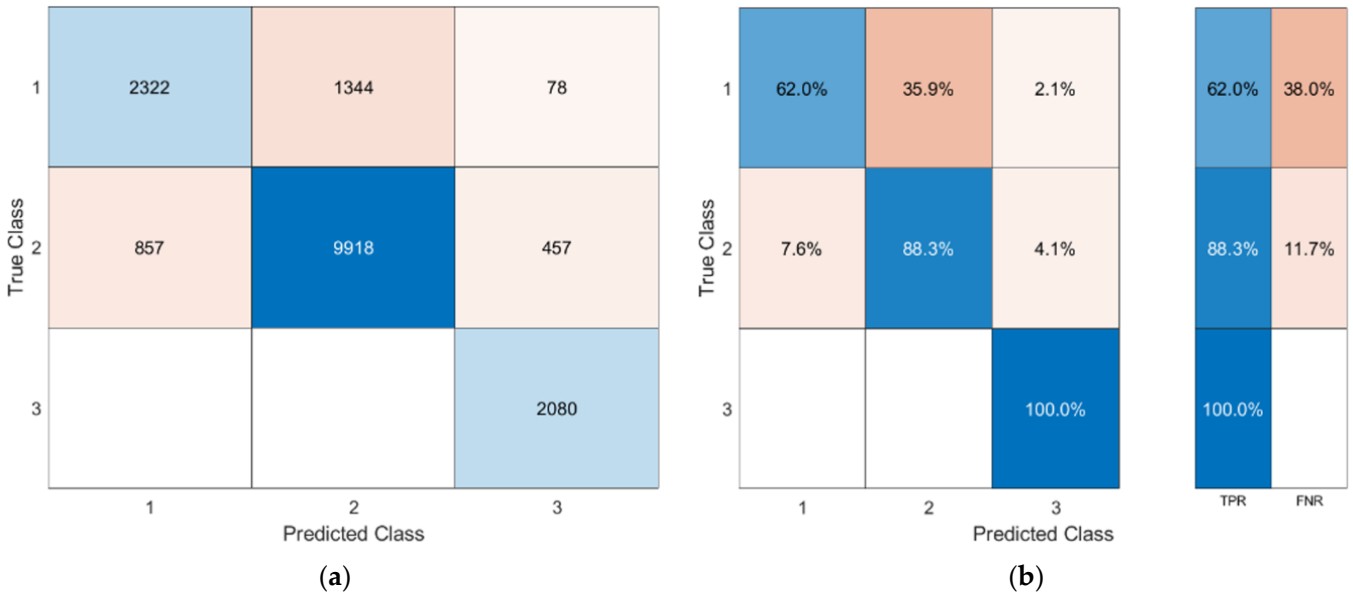

**Figure 6.** Confusion matrix corresponding to the highest accuracy shown in Table 6: (**a**) confusion matrix showing numerical values (**b**) TPR and FNR (in percentages).

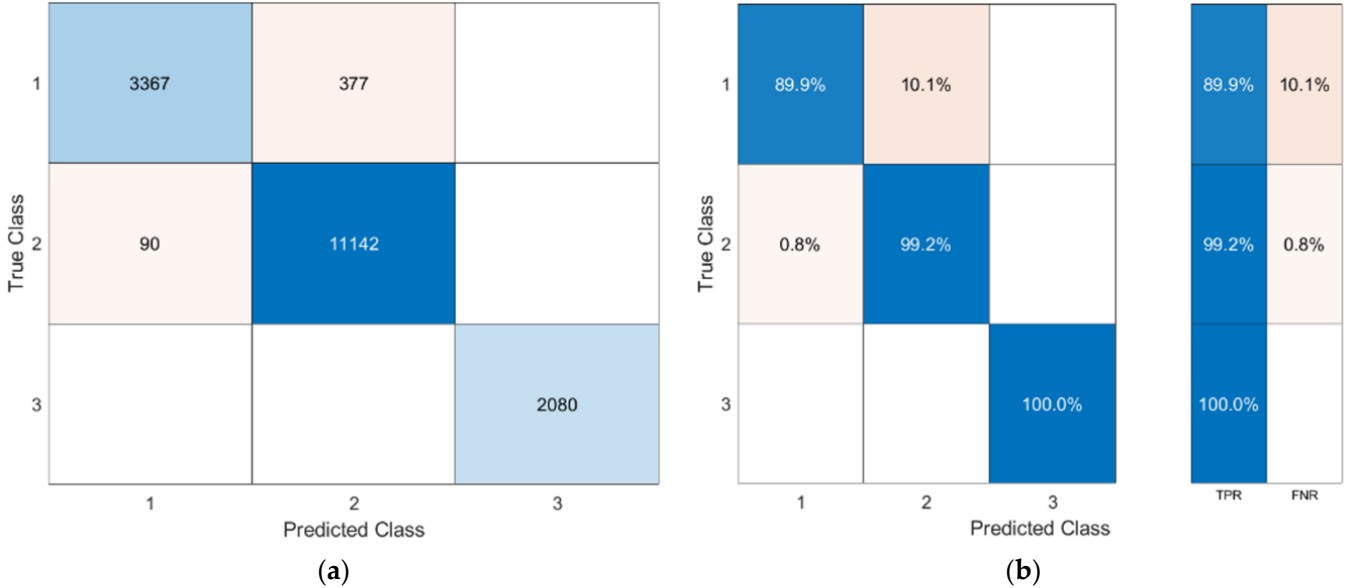

**Figure 7.** Confusion matrix corresponding to the highest accuracy shown in Table 7: (**a**) confusion matrix showing numerical values (**b**) TPR and FNR (in percentages).

**Table 10.** Performance of data-mining techniques in fault detection with noisy conditions.

| Data-Mining Methods | Selected Features | Accuracy (No Noise) | Accuracy (20 dB Noise) | Avg. Testing Time |
|---|---|---|---|---|
| ANN | F11,F14,F3,F7,F8,F5,F12,F10,F6,F13,F9,F1 | 91.5% | 82.5% | 8.05 ms |
| SVM | F3,F2,F8,F12,F6,F1,F10,F5,F11,F4,F7,F13,F14,F9 | 97.3 | 88.6% | 6.03 ms |
| ELM | F3,F8,F11,F13,F6,F10,F2,F14,F5,F9,F12,F1 | 84% | 81.3% | 1.59 ms |
| RF | F4,F3,F1,F2,F7,F5,F6,F8,F9,F10,F11,F12,F13,F14 | 96.7% | 88.7% | 0.58 ms |
| KNN | F3,F12,F6,F4,F5,F10,F13,F14,F1,F2 | 98.8% | 92.0% | 0.89 ms |
| DBN | F1,F2,F3,F4,F5,F6,F7,F8,F9,F10,F11,F12,F13,F14 | 98.9% | 91.8% | 2.59 ms |

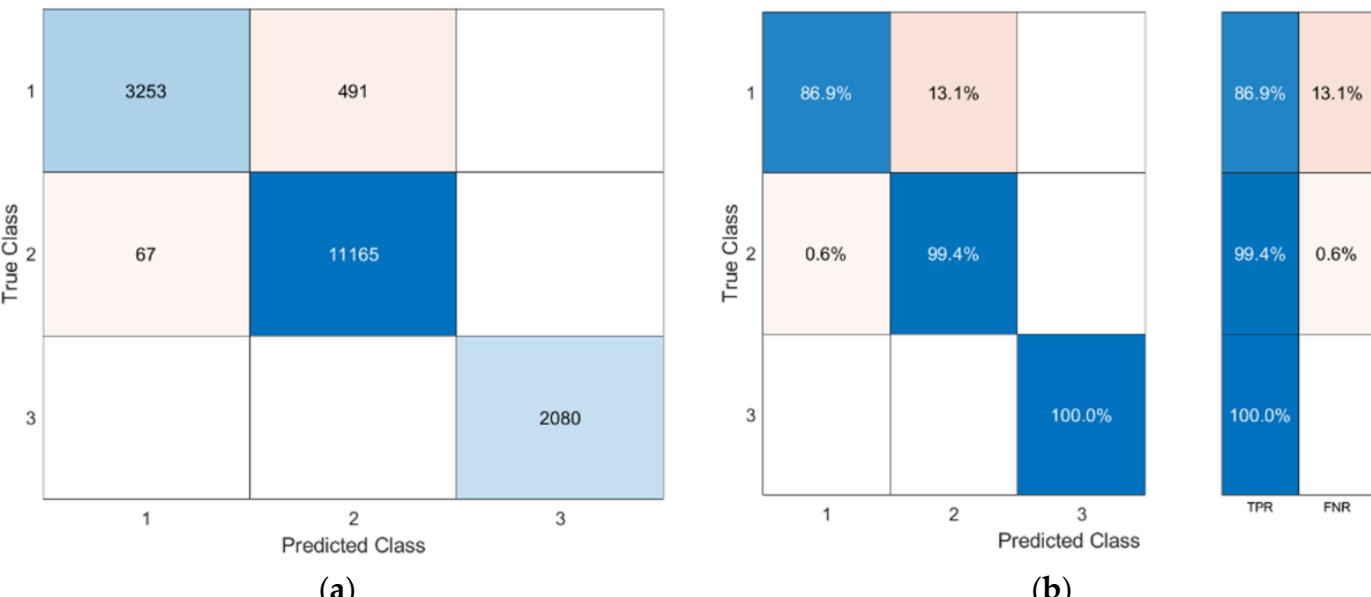

**Figure 8.** Confusion matrix corresponding to the highest accuracy shown in Table 8: (**a**) confusion matrix showing numerical values (**b**) TPR and FNR (in percentages).

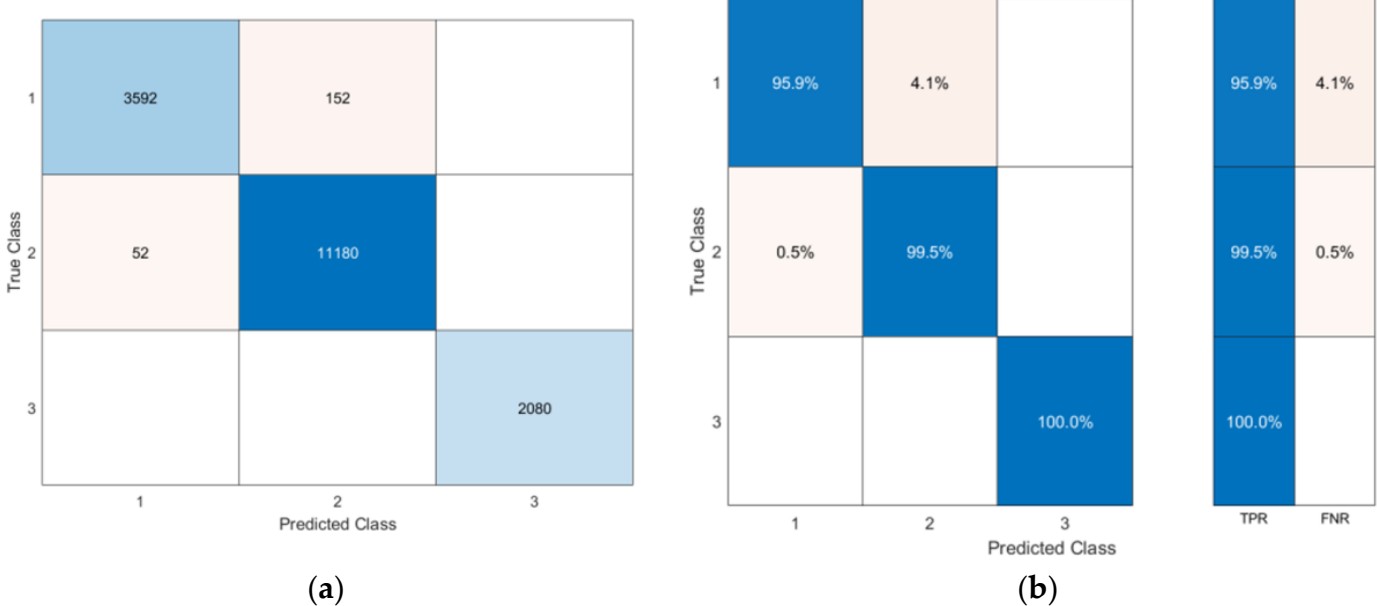

**Figure 9.** Confusion matrix corresponding to the highest accuracy shown in Table 9: (**a**) confusion matrix showing numerical values (**b**) TPR and FNR (in percentages).

### 5.2. Fault Location

As presented in Section 3.2, the DC fault events were simulated in the considered HVDC systems with varying fault resistances and locations. After fault events were classified as DC internal faults, the accurate detection of fault location was considered as an important task of the protection system. In this work, a fault location algorithm was also designed for the protection system, as shown in Figure 2. Similar to the fault classification task, the location of the fault was accomplished through the training and testing of different data regression techniques based on data-mining approaches (such as ANN, SVR, ELM, LR, RF, and DBN). To test the performance of each DMT, several performance indices or error indices (mathematically represented in Equations (9)–(13) were calculated and analyzed. The mean square error (MSE) and root-mean-square error

(RMSE) were the two most familiar performance indicators which generally certified how close the predicted values were to real values. Here, a smaller error signified that the predicted value was closer to the actual one. The normalized RMSE (NRMSE) provided a comparative assessment between models with different scales. Similar to MSE and RMSE, a smaller value of calculated MAPE showed how close the forecast values were to the actual value. The scale-independencies and interpretability were two major advantages of MAPE. As compared with other performance indicators, a higher rate (maximum '1') of R-value indicated how well the prediction was achieved by the model.

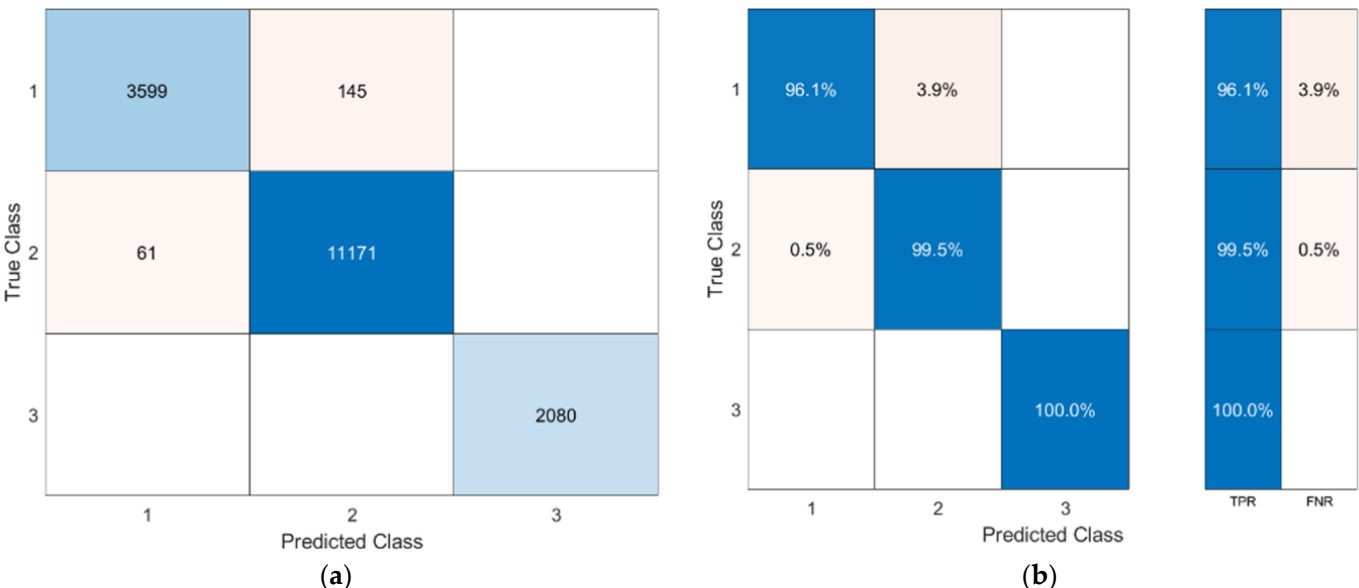

**Figure 10.** Confusion matrix corresponding to DBN: (**a**) confusion matrix showing numerical values (**b**) TPR and FNR (in percentages).

$$\text{Mean Square Error (MSE)} = \frac{1}{k} \sum_{i=1}^{k} (AV_i - PV_i)^2 \tag{9}$$

$$\text{R} - \text{Value} = 1 - \left( \left| \frac{\sum_{i}^{k} PV_i}{\sum_{i}^{k} AV_i} \right| \right) \tag{10}$$

$$\text{Root Mean Square Error (RMSE)} = \sqrt{\frac{\sum_{i=1}^{k} \left\| (AV_i - PV_i)^2 \right\|}{k}} \tag{11}$$

$$\text{Normalized Root Mean Square Error(NRMSE)} = \frac{\sqrt{\sum_{i=1}^{k} \left\| (AV_i - PV_i)^2 \right\|} / k}{AV_{\max} - AV_{\min}} \tag{12}$$

$$\text{Mean absolute percentage error (MAPE)} = \frac{1}{k} \sum_{i=1}^{k} \left\| \frac{(AV_i - PV_i)}{AV_i} \right\| \tag{13}$$

where '*k*' signifies the number of samples. *AV* and *PV* signify the actual value and predicted value, respectively.

All the simulated internal DC faults as signals (3744 numbers, as presented in Section 2) were trained and tested by the DMTs with all 14 features as input data and fault location (or distance) from the respective relay points as target data. The error indices for each DMT are

presented in Table 11. As presented in Table 11, the SVR-, ELM-, and LR-based regression techniques were found to be worst compared with other DMTs. The studied performance indicators of SVR, ELM, and LR were found, respectively, as: MSE: 654.7105, 591.977, and 595.2575; RMSE: 25.5872, 22.3306, and 24.3979; NRMSE: 0.3198, 0.3041, and 0.3050; MAPE: 75.2735, 76.5244, and 75.9142; R-Value: 0.7932, 0.8131, and 0.8120. However, the ANN and KNN techniques showed an improved performance compared with SVR and ELM. For example, the MSE of ANN and KNN were found to be 75.6485 and 65.0271 compared with SVR (654.71) and ELM (591.97), which was an 8–10 times lesser error of prediction.

**Table 11.** Performance of data-mining techniques in fault location.

| DMTs | MSE | R-Value | RMSE | NRMSE | MAPE |
|------|------|---------|------|-------|------|
| ANN | 75.6485 | 0.9721 | 8.6976 | 0.1087 | 18.4687 |
| SVR | 654.7105 | 0.7932 | 25.5872 | 0.3198 | 75.2735 |
| ELM | 591.9770 | 0.8131 | 22.3306 | 0.3041 | 76.5244 |
| KNN | 65.0271 | 0.9795 | 8.0639 | 0.1008 | 17.7980 |
| LR | 595.2575 | 0.8120 | 24.3979 | 0.3050 | 75.9142 |
| RF | 2.2737 | 0.9993 | 1.5079 | 0.0188 | 2.6403 |
| DBN | 2.1116 | 0.9993 | 1.4531 | 0.0182 | 2.7047 |

The RF showed a significantly better result with the lowest error rate (such as MSE = 2.2737, RMSE = 1.5079, NRMSE = 0.0188, and MAPE = 2.6403) compared with other ML-based DMTs. The R-value value for RF was noted as 0.9993 (very close to 'one'). However, DBN (a DL-based DMT) achieved a similar kind of result as RF achieved. The output performance indices of DBN were as follows: MSE = 2.1116, RMSE = 1.4531, NRMSE = 0.0182, MAPE = 2.7047, and R-value= 0.9993. A sample result corresponding to fault location with different fault locations are shown in Table 12.

**Table 12.** Sample results corresponding to fault location occurred at Line-1 with $R_f$ = 0.02 $\Omega$ι.

| Data-Mining Methods | Actual Length | Predicted Length | Absolute Error (%) | Actual Length | Predicted Length | Absolute Error (%) | Actual Length | Predicted Length | Absolute Error (%) |
|---------------------|---------------|------------------|--------------------|---------------|------------------|--------------------|---------------|------------------|--------------------|
| ANN | 30 | 26.8046 | 10.65 | 60 | 64.7104 | 7.85 | 90 | 95.8205 | 6.46 |
| ELM | 30 | 38.7092 | 29.03 | 60 | 44.4856 | 25.85 | 90 | 56.5676 | 37.14 |
| SVR | 30 | 39.6720 | 32.24 | 60 | 42.1258 | 29.79 | 90 | 63.0163 | 29.98 |
| KNN | 30 | 27.0201 | 9.93 | 60 | 56.2587 | 6.23 | 90 | 86.00 | 4.44 |
| LR | 30 | 37.5278 | 25.09 | 60 | 46.5935 | 22.34 | 90 | 68.4625 | 23.9 |
| RF | 30 | 30.3111 | 1.037 | 60 | 58.2687 | 2.88 | 90 | 89.6505 | 0.38 |
| DNN | 30 | 30.1025 | 0.34 | 60 | 59.0548 | 1.57 | 90 | 90.4568 | 0.50 |

## 6. Conclusions

In this work, the fault detection and location in a VSC-HVDC system are studied by the use of data-mining techniques. Several internal (DC) and external (both DC and AC sides) are simulated with possible system conditions and fault situations. In the beginning, the DC-voltage and DC-current signals are retrieved at relay terminals of the studied HVDC system, which are later used to extract fourteen sensitive features. After that, a black-box solution-based data-mining approach is presented in order to detect and classify the faults. In the DMT framework, several approaches are tested such as ANN, SVM, KNN, ELM, LR, RF, KNN, and DBN. In order to increase the efficiency and reduce the computational burden of the data-mining model-based fault classifier, the sequential forward feature selection is integrated with each DM model. It has been studied that the KNN classifier as an ML-based

DMT and DBN as a based DMT perform extremely well to classify the external and internal faults in the HVDC system. The performance (accuracy and dependability) of KNN and DBN are noted to be ~99% and ~96%. In the 20 dB noisy environment, the classifiers show an accuracy of ~92%. After fault events are classified as DC internal faults, the accurate fault location is carried out with different DMTs. As compared with other DMTs, the RF- and DBN-based regressors perform better in order to predict the location of faults. The DBN predicts the fault location with the least error (MSE = 2.116 and RMSE = 1.4531).

Testing the proposed approach in a renewable energy-penetrated VSC-HVDC system can be considered as a future scope of the work. Moreover, in the data-mining framework, other advanced deep learning networks such as long–short-term-memory networks, convolution neural networks, and transfer learning schemes can be implemented, which gives scope for future research.

**Author Contributions:** Conceptualization, M.M. and D.A.G.; methodology, M.M.; software, A.P.; validation, A.P., D.A.G. and T.P.; formal analysis, M.M.; investigation, A.P.; resources, T.P.; data curation, D.A.G.; writing—original draft preparation, A.P. and M.M.; writing—review and editing, A.P., D.A.G. and M.M.; visualization, M.M.; supervision, M.M. and D.A.G. All authors have read and agreed to the published version of the manuscript.

**Funding:** This research received no external funding.

**Data Availability Statement:** Not applicable.

**Acknowledgments:** We acknowledge the Siksha O Anusandhan University, Bhubaneswar, for providing the lab facilities for conducting this work.

**Conflicts of Interest:** The authors declare no conflict of interest.

## Abbreviations

The following abbreviations are used in the manuscript:

| | |
|---|---|
| VSC | voltage source converter |
| HVDC | high voltage direct current |
| ML | machine learning |
| DL | deep learning |
| DBN | deep belief network |
| DMT | data-mining techniques |
| HVAC | high voltage alternating current |
| TW | traveling wave |
| ANN | artificial neural network |
| AI | artificial intelligence |
| SVM | support vector machine |
| ELM | extreme learning machine |
| KNN | k-nearest neighbor |
| RF | random forest |
| SFS | sequential forward feature selection |
| LR | linear regression |
| P-G | positive pole-to-ground |
| N-G | negative pole-to-ground |
| P-P | pole-to-pole |
| P-P-G | pole-to-pole-to-ground |
| $R_{fault}$ | fault resistances |
| $L_f$ | fault location |
| $F_T$ | fault type |
| DT | decision tree |
| RBM | restricted Boltzmann machines |
| TPR | true positive rate |
| FNR | false negative rate |
| MSE | mean square error |

| | |
|---|---|
| RMSE | root-mean square error |
| NRMSE | normalized root-mean square error |
| MAPE | mean absolute percentage error |
| $m$ | weight |
| $b$ | bias |
| $\zeta$ | error |
| $P$ | penalty factor |
| $T$ | targeted variable |
| $\alpha_1$ | linear regression coefficient |
| $e$ | random error |
| $k$ | number of samples |
| $AV$ | actual value |
| $PV$ | predicted value |

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
