# Peer review of "Data-Mining Techniques Based Relaying Support for Symmetric-Monopolar-Multi-Terminal VSC-HVDC System"

_asi, doi:10.3390/asi6010024_

Round 1
Reviewer 1 Report
The topic seems to be interesting for readers. However, this review could not follow the main contribution of the conducted study. In this round of review followings should be addressed.
1. The presentation quality is not satisfactory. Major revision is recommended to enhance it.
To do so,
2. Abstract and intro should be revised.
3. Proper literature review is recommended.
4. Figures are not clear. Revise them.
5. Paper structure also needs revision.
Author Response
Response to and Reviewers’ Comments
The authors would like to thank the Editor and Reviewers for the time they spent reviewing our manuscript ID: asi-2185736 entitled “Data-Mining techniques based relaying support for Symmetric-Monopolar-Multi-terminal VSC-HVDC system” submitted to the esteemed journal: Applied System Innovation. The suggestions provided were constructive to enhance the quality of the manuscript. We have tried to answer all your questions/remarks/suggestions to the best of our knowledge. We hope you find the answers satisfactory and the manuscript of better quality.
To help the editor and reviewers to find the modifications introduced in the revised manuscript, the following set of color codes is used.
- Red text: Comments of the Editor and Reviewers in the response sheet.
- Black text: Authors’ replies to the comments of the Editor and Reviewers in the response sheet.
- Yellow patch: Revisions made in the manuscript file
Reviewer-1 Comments
The topic seems to be interesting for readers. However, this review could not follow the main contribution of the conducted study. In this round of review followings should be addressed.
Response: Thank you for appreciating our work and your valuable suggestions. The contribution of the study is highlighted with bullet points in the introduction section. We have considered all of your suggestions in our revised manuscript.
Comment 1. The presentation quality is not satisfactory. Major revision is recommended to enhance it.
Response: Thanks for the suggestion. With respect to the reviewer's concern, we have tried our level best to improvise the presentation quality in the revised manuscript.
Comment 2. Abstract and intro should be revised.
Response: Thanks to the reviewer for this suggestion. Considering the reviewer's suggestion, the abstract section is re-written in such a way that it can able to provide information about the major objective, contribution, and major outcomes.
Similarly, we have considered the suggestion of the reviewer to revise the introduction section. In this regard, the introduction section is subdivided into (1.1) Motivation and incitement, (1.2) Literature review, and (1.3) Contribution and Organization.
The modification in the abstract and introduction section is highlighted in yellow marker.
Comment 3. Proper literature review is recommended.
Response: Thanks for the comment. In the revised manuscript, the literature review section (section 1.2) has been completely rephrased and reorganized with a proper justification for the presented study, motivation, and objective. The respective changes are highlighted in Section 1.2 and in the updated references section.
Comment 4. Figures are not clear. Revise them.
Response: Thanks to the reviewer for this observation. Considering the reviewer's suggestion, the figures are replaced with clearly visible figures in the revised manuscript with better quality having a resolution of more than 600dpi.Moreover, we have uploaded a new file having all the figures with 600dpi resolution in the submission system.
Comment 5. Paper structure also needs revision.
Response: Thanks for the suggestion to improve the paper. We have tried to improvise the main structure of the paper with respect to this Journal-Standard. The main structure of the paper is as follows:
1. Introduction |
|
|
1.1. Motivation and Incitement |
|
1.2. Literature review |
|
1.3. Major Contribution and Organization |
2. Studied HVDC System |
|
3. Methodology |
|
|
3.1. Fault and No-fault Data generation |
|
3.2. Feature extraction |
4. Studied DMTs for HVDC fault recognition |
|
|
4.1. Artificial Neural Network (ANN) |
|
4.2. Support Vector Machine (SVM) |
|
4.3. Support Vector Regression (SVR) |
|
4.4. Extreme Learning Machine (ELM) |
|
4.5. K-Nearest Neighbour (KNN) |
|
4.6. Random Forest Algorithm (RF) |
|
4.7. Linear Regression Algorithm (LR) |
|
4.8. Deep Belief Network (DBN) |
5. Result and Discussion |
|
|
5.1 Fault events classification |
|
5.2. Fault Location |
6. Conclusions |
|
Nomenclature |
|
References |
|
--------------------------------------------------------------------------------------------------------------------------
Thank you again for reviewing our paper and providing valuable suggestions to improve the quality of the paper.

Reviewer 2 Report
In fact, the paper deals with a good topic in protection of HVDC transmission systems based on data mining. In my opinion, the paper can be accepted for publication after carrying out the following comments:
1- English language should be improved as there are several grammatical errors.
2- The methodology section should be extremely improved.
3- The flowchart of figure 2 should be described in details in the paper text.
4- Each table should be described and cited in the paper text, for example table 1.
5- Some sentences should be completed. For example, "13 fault resistances ........." in subsection 3.1.
6- It is better to give more discussions about the differences obtained in fault location accuracy of the adopted different techniques.
7- It will be better if the authors apply this study on HVDC system containing renewable energy resources such as wind and PV systems.
Author Response
Response to and Reviewers’ Comments
The authors would like to thank the Editor and Reviewers for the time they spent reviewing our manuscript ID: asi-2185736 entitled “Data-Mining techniques based relaying support for Symmetric-Monopolar-Multi-terminal VSC-HVDC system” submitted to the esteemed journal: Applied System Innovation. The suggestions provided were constructive to enhance the quality of the manuscript. We have tried to answer all your questions/remarks/suggestions to the best of our knowledge. We hope you find the answers satisfactory and the manuscript of better quality.
To help the editor and reviewers to find the modifications introduced in the revised manuscript, the following set of color codes is used.
- Red text: Comments of the Editor and Reviewers in the response sheet.
- Black text: Authors’ replies to the comments of the Editor and Reviewers in the response sheet.
- Yellow patch: Revisions made in the manuscript file
Reviewer-2 Comments
In fact, the paper deals with a good topic in protection of HVDC transmission systems based on data mining. In my opinion, the paper can be accepted for publication after carrying out the following comments:
Response: Thank you for appreciating our work and your valuable suggestions. We have considered all of your suggestions in our revised manuscript.
Comment 1. English language should be improved as there are several grammatical errors.
Response: Thanks to the reviewer for this observation. The English have been checked thoroughly online by the Grammarly software and necessary improvement has been done. Moreover, the revised manuscript has been re-checked by a senior professor of the English language in order to make it error-free. The corrections are highlighted in green colour text. Moreover, an additional Abbreviation section is inserted for the betterment of the paper.
Comment 2. The methodology section should be extremely improved.
Response: Thanks to the reviewer for this suggestion. We have tried our level best to improve the writing of the methodology section (Section 3)in the revised manuscript. The changes are highlighted with yellow marker in the revised manuscript (first paragraph of Section 3, Bullet points in section 3.1, and last paragraph of section 3.2).
Comment 3. The flowchart of figure 2 should be described in details in the paper text.
Response: Thanks for the comment. Considering the reviewer suggestion, the figure-2 is comprehensively described in the revised manuscript. The changes are highlighted with yellow marker in the revised manuscript ( first paragraph of Section 3).
Comment 4. Each table should be described and cited in the paper text, for example table 1.
Response: Thank you for your concern. The Table 1 has been cited properly in the revised manuscript (Section-2,2nd paragraph: last line). Moreover, we have rechecked the manuscript for avoiding similar types of errors.
Comment 5. Some sentences should be completed. For example, "13 fault resistances ........." in subsection 3.1.
Response: Thanks for the suggestion to improve the paper. The lines are modified with proper meaning in the revised manuscript. The changes are highlighted with yellow marker in the revised manuscript (Section 3.1). Moreover, we have rechecked the manuscript for avoiding similar types of errors.
Comment 6. It is better to give more discussions about the differences obtained in fault location accuracy of the adopted different techniques.
Response: Thanks for the suggestion to improve the paper. Considering the reviewer's suggestion, Section 3.2 which provides the result analysis of fault location is extensively amended in the revised manuscript. The significance of each performance indicator is comprehensively described with clear analysis. Moreover, the differences obtained in fault location accuracy of the adopted different techniques are compared with respect to the studied performance indicators. The respective changes are highlighted with yellow markers in Section 3.2 (1st, 2nd, and 3rd paragraph).
Comment 7. It will be better if the authors apply this study on HVDC system containing renewable energy resources such as wind and PV systems.
Response: Thanks for the suggestion to improve the paper. We would like to inform you that we have already started the simulation work for the HVDC system integrated with renewable energy sources (mainly the wind turbine system). Generating several thousands of data considering each diverse operating condition is taking a lot of time; therefore, it will be quite impossible for us to apply in the present paper. However, we will definitely work with the reviewer's suggested renewable energy-integrated HVDC system in our future work. Concerning the reviewer's valuable suggestions, the conclusion section is amended with a few sentences highlighting the future scope of the studies.
---------------------------------------------------------------------------------------------------------------------------
Thank you again for reviewing our paper and providing valuable suggestions to improve the quality of the paper.

Round 2
Reviewer 2 Report
The authors took most of the comments into consideration. In my opinion, the paper can be accepted in present form.